# An imaging flow cytometry-based methodology for the analysis of single extracellular vesicles in unprocessed human plasma

Wouter W. Woud [1✉], Edwin van der Pol [2,3], Erik Mul[4], Martin J. Hoogduijn[1], Carla C. Baan[1], Karin Boer [1] & Ana Merino[1]

Extracellular vesicles (EVs) are tissue-specific particles released by cells containing valuable diagnostic information in the form of various biomolecules. To rule out selection bias or introduction of artefacts caused by EV isolation techniques, we present a clinically feasible, imaging flow cytometry (IFCM)–based methodology to phenotype and determine the concentration of EVs with a diameter ≤400 nm in human platelet-poor plasma (PPP) *without* prior isolation of EVs. Instrument calibration (both size and fluorescence) were performed with commercial polystyrene beads. Detergent treatment of EVs was performed to discriminate true vesicular events from artefacts. Using a combination of markers (CFSE & Tetraspanins, or CD9 & CD31) we found that >90% of double-positive fluorescent events represented single EVs. Through this work, we provide a framework that will allow the application of IFCM for EV analysis in peripheral blood plasma in a plethora of experimental and potentially diagnostic settings. Additionally, this direct approach for EV analysis will enable researchers to explore corners of EVs as cellular messengers in healthy and pathological conditions.

---

[1] Erasmus MC Transplant Institute, Department of Internal Medicine, University Medical Center Rotterdam, Rotterdam, The Netherlands. [2] Biomedical Engineering & Physics, Laboratory Experimental Clinical Chemistry, Vesicle Observation Center, Amsterdam UMC, University of Amsterdam, Amsterdam, The Netherlands. [3] Cancer Center Amsterdam, Imaging and Biomarkers, Amsterdam, The Netherlands. [4] Department Central Cell Analysis Facility, Sanquin Research and Landsteiner Laboratory, Academic Medical Center, University of Amsterdam, Amsterdam, The Netherlands. ✉email: w.woud@erasmusmc.nl

Extracellular vesicles (EVs) are lipid bilayer membrane structures (30–8000 nm in diameter[1]) released by cells. They are involved in cellular communication through transfer of surface receptors and/or a variety of macromolecules carried as cargo (e.g., lipids, proteins, nucleic acids, protein-coding mRNAs and regulatory microRNAs)[2,3]. As EVs are excreted by virtually all cell types in the human body, they can be found in most body fluids, such as the blood[1], saliva[4] and urine[5,6]. Often regarded as a "snapshot" of the status of the cell of origin, EVs are examined for their biochemical signatures to assess the presence of various diseases, e.g., cancer or viral infections[7,8], and are considered excellent minimally invasive biomarkers in so-called liquid biopsies[9–11]. While no unique antigens representative for specific EV classes and subpopulations have been reported to date, tetraspanins (CD9/CD63/CD81) are recognized as common antigens. These proteins are enriched on EVs and are involved in EV biogenesis, cargo selection, and cell targeting[12,13].

Despite the increased interest in EVs as biomarker, their quantification and characterization is hampered by physical characteristics such as their small size and low epitope copy number[14], the variety of their protein markers depending on the cell source, and the confinement of some markers to the luminal side of the EVs[3,15]. The identification of EVs in blood plasma is further hindered by the molecular complexity of the plasma, which contains multiple elements (e.g., lipoproteins, cell debris and soluble proteins), that interfere with EV analysis[3,16]. Moreover, a lack of robust methods and ambiguities in how data should be interpreted for EV analysis makes data interpretation between studies challenging[17,18].

Currently, the gold standard approach for EV analysis is based on the isolation or concentration of EVs. Ultracentrifugation, density-gradient, and size exclusion chromatography are the most widely used EV isolation techniques[19], despite yielding low-purity EV samples due to the co-isolation of non-desired molecules such as lipoproteins[3,16]. Additionally, a variety of analytical platforms are available. Nanoparticle tracking analysis (NTA) allows the determination of the size distribution and a rough indication of the concentration[20] of individual nanoparticles in suspension, but provides limited phenotyping capabilities. In turn, transmission electron microscopy (TEM) is able to image particles <1 nm, but is time consuming. Other methods, such as ELISA and Western blot analysis, offer bulk phenotyping abilities but lack quantification[5,21–23]. Thus, a tool for the accurate determination of the concentration and phenotyping of single EVs in complex samples such as plasma represents an unmet need.

Flow Cytometry (FC) is a tool to quantify and phenotype particles in suspension. However, while EVs can reach sizes up to ~8000 nm in diameter, the majority of EVs are <300 nm and are therefore difficult to discriminate from background noise by conventional FC[3,24,25]. In recent years, imaging flow cytometry (IFCM) has emerged as a technique that enables the discrimination and analysis of single EV. The ability of IFCM to detect submicron particles has been demonstrated by several research groups using fluorescent polystyrene beads[26–29] or the use of cell supernatant-derived EVs[21]. To date, several studies have reported the detection of EVs - obtained after performing isolation procedures - from plasma using IFCM[26,27,29,30]. However, due to the used isolation procedures, it is difficult to evaluate whether these results represent all EVs in plasma, or if some subpopulations are missed[31].

To rule out selection bias or introduction of artefacts caused by EV isolation techniques, we here demonstrate an IFCM-based methodology to phenotype and determine the concentration of human plasma-derived EVs with a diameter ≤400 nm - *without* prior isolation of EVs. By omitting the need for sample isolation,

this method is able to directly show the status of an individual, which will be greatly beneficial in the monitoring of EVs in health and disease, and will enable researchers to explore corners of EV biology.

## Results

**Outline of the article**. The objective of this article is to provide an assay that will allow researchers to study single EVs directly in diluted, labeled human plasma using IFCM. The following procedures were conducted to validate our assay: size calibration of the IFCM based on scatter intensities, background analysis of the IFCM, detergent treatment of EVs, dilution experiments, and fluorescence calibration. In addition, two labeling strategies based on CFSE + Tetraspanin+ and CD9 + CD31 + were evaluated by mixing human plasma with mouse plasma at different ratios.

**Detection of sub-micron fluorescent polystyrene beads**. EV analysis at the single EV level requires an instrument that is able to detect a heterogeneous submicron-sized population. To this end, we tested the ability of IFCM to discriminate single-size populations of fluorescent sub-micron beads by measuring two commercially available mixtures of FITC-fluorescent polystyrene (PS) beads of known sizes (Megamix-Plus FSC – 900, 500, 300 and 100 nm, and Megamix-Plus SSC – 500, 240, 200, 160 nm). Within the Megamix-Plus FSC mix, we acquired a 300/500 nm bead ratio of 2.2, which is within the manufacturers internal reference qualification range (1.7–2.7 ratio). Next, we mixed both bead sets in a 1:1 ratio ('Gigamix') and performed acquisition. Figure 1a shows that IFCM is able to discern all seven fluorescent bead populations, as well as the 1 μm-sized Speed Beads (SB), via the FITC (Ch02) and side scatter (SSC - Ch06) intensities.

**Calibration of scatter intensities through Mie theory**. The output of IFCM signal intensities are presented in arbitrary units (a.u.), which hinders data comparability (and reproducibility) with different flow cytometers. Since light scattering of spherical objects is dependent on particle size and refractive index, Mie theory can be used to relate the scatter intensity of events to their size given their refractive index[32]. Generally, Mie theory is applied to calibrate the scatter channels of a FC (forward- and/or sideward-scattered light - FSC or SSC, respectively); however, IFCM utilizes a brightfield detection channel (BF, Ch04) as opposed to FSC.

Mie theory was applied on both scatter detection channels (BF and SSC). As a first step, we extracted the BF and SSC median scatter intensities of each identified size population of PS beads (Fig. 1b). Coefficient of variation (CV) analysis for each single PS bead population showed scores ≥8% for the BF detector irrespective of bead size, whereas CV scores for the SSC detection channel were observed to increase with decreasing bead sizes – indicating that the detection of smaller particles is close to the detection limit of the SSC detector in our setup.

Next, BF and SSC data of the PS beads were scaled onto Mie theory, resulting in a scaling factor (F) of 1.3518 and a coefficient of determination ($R^2$) of 0.00 for the BF detector and a scaling factor of 8.405 and an $R^2$ of 0.91 for the SSC detector (Fig. 1c). Thus, signals from sub-micron PS beads measured with the BF detector do not provide quantitative information. The SSC detector, on the other hand, can be readily calibrated. For the SSC detector, the theoretical model indicates a plateau for EVs with a diameter between ~400 and ~800 nm, which translates into a low resolution when determining EV sizes based on SSC intensities within this region. To ensure inclusion of sub-micron EVs, a gate was set at SSC below the scattering intensity

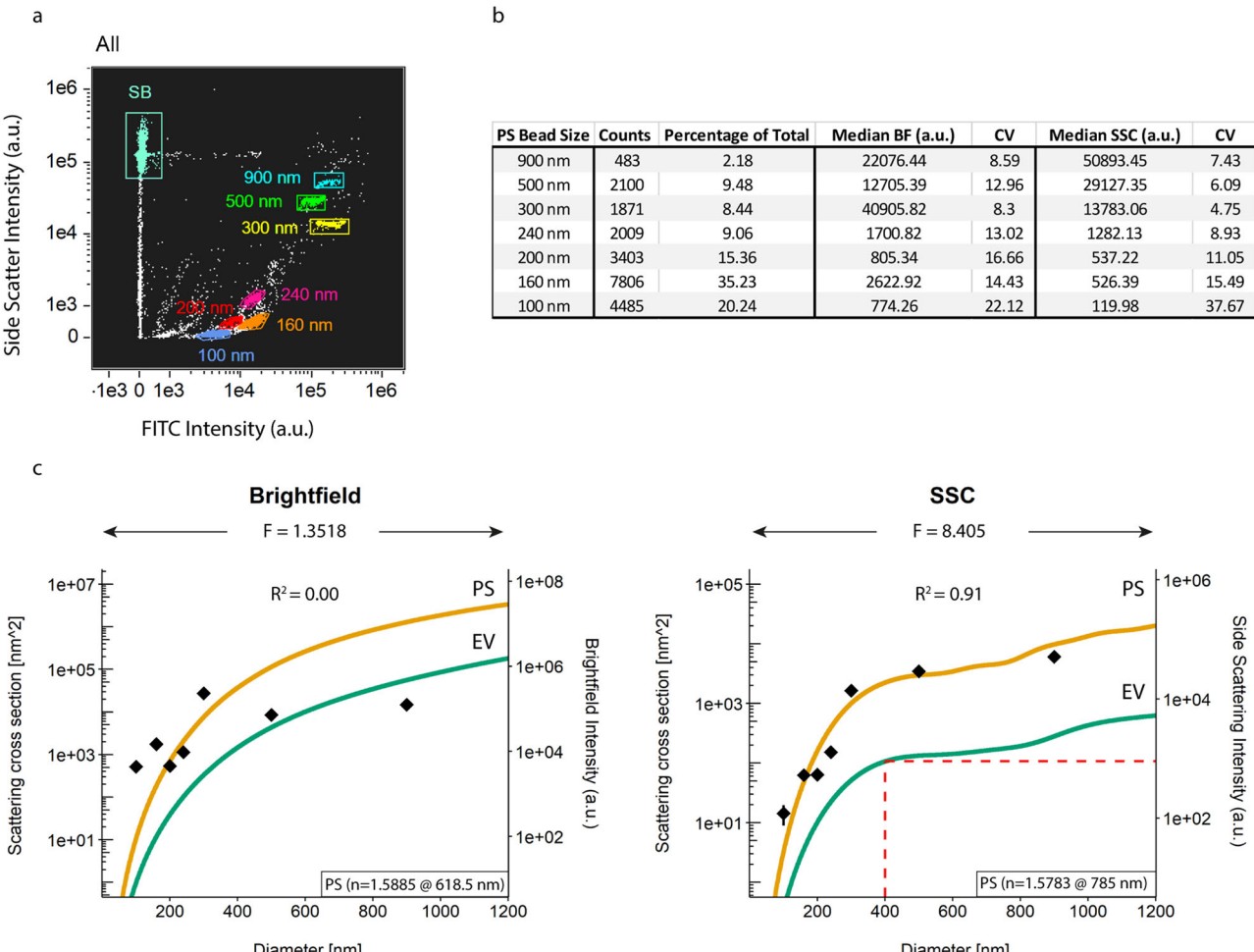

**Fig. 1 Calibration of scatter intensities through Mie theory. a** Gigamix polystyrene (PS) bead populations with sizes from 900 nm down to 100 nm were identified on the basis of SSC and FITC fluorescent intensities. **b** Counts and median scatter intensities of each PS bead population as detected by the brightfield (BF) and side scatter (SSC) detectors (Ch04 and Ch06, respectively). **c** Diameter vs Scattering cross section graphs. PS beads (green lines) were modelled as solid spheres with a refractive index of 1.5885 for a wavelength of 618.5 nm (brightfield) and 1.5783 for a wavelength of 785.0 nm (SSC). EVs (orange lines) were modelled as core-shell particles, with a core refractive index of 1.38 and a shell refractive index of 1.48 and a shell thickness of 6 nm for both wavelengths. The obtained scatter intensities of the PS beads as described in **b** were overlayed and a least-square-fit was performed to correlate theory and practice. Based on these correlations, SSC signal intensities were found to be indicative of particle size and a SSC cut-off of 900 a.u – corresponding to particles of 400 nm – was used in the rest in this work. F: scaling factor between scattering intensity and scattering cross section; n: refractive index.

corresponding to the plateau, namely 400 nm EVs, corresponding to a value of 900 a.u. SSC intensity.

These data show that 1) IFCM is able to readily discern submicron-sized EVs based on their emitted fluorescence and SSC intensities, and 2) SSC – but not BF – light scattering intensities can be used to approximate particle sizes (following Mie calculations). The standardization of SSC signal intensities followed by the setting of a sub-micron gate provides a tool to selectively analyze all fluorescent EVs in complex samples such as plasma, as long as these particles emit detectable fluorescent intensities.

**IFCM gating strategy for the detection of single particles ≤ 400 nm in plasma.** EVs represent a heterogeneous group with different cellular origin. The analysis of single EVs, as well as the different subsets, will provide a better understanding of the pathophysiological state of the individual. Therefore, we designed a gating strategy to analyze individual submicron-sized particles based on 1) the analysis of events within a predefined submicron size range, and 2) exclusion of multispot fluorescent events from our analysis.

Based on the previous results, we selected all events with SSC intensities ≤900 a.u. - corresponding with particles of 400 nm and below. (Fig. 2a–I). Next, we checked for multiplet detection within each separate fluorescent detection channel based on the number of fluorescent spots within the pixel grid for each acquired event: these spots were quantified by combining the "Spot Count" feature with the intensity masks for each of the channels used per experiment. Although the camera can spatially resolve signals originating from multiple simultaneously imaged EVs, the software anticipates that the signals are originating from multiple locations within 1 cell. By selecting all events that showed 0 or 1 spot, representing either negative or single-positive events for a fluorescent marker, we were able to exclude multiplet events from our analysis (Fig. 2a – II, III). As a last step, we calculated the distance between individual fluorescent spots detected in different fluorescent channels to exclude any false double-positive events (defined as 2 different single-positive particles within the same event). To this end, we created a new mask by combining the intensity masks of the channels in use per experiment using Boolean logic (e.g., MC_Ch02 OR MC_Ch05),

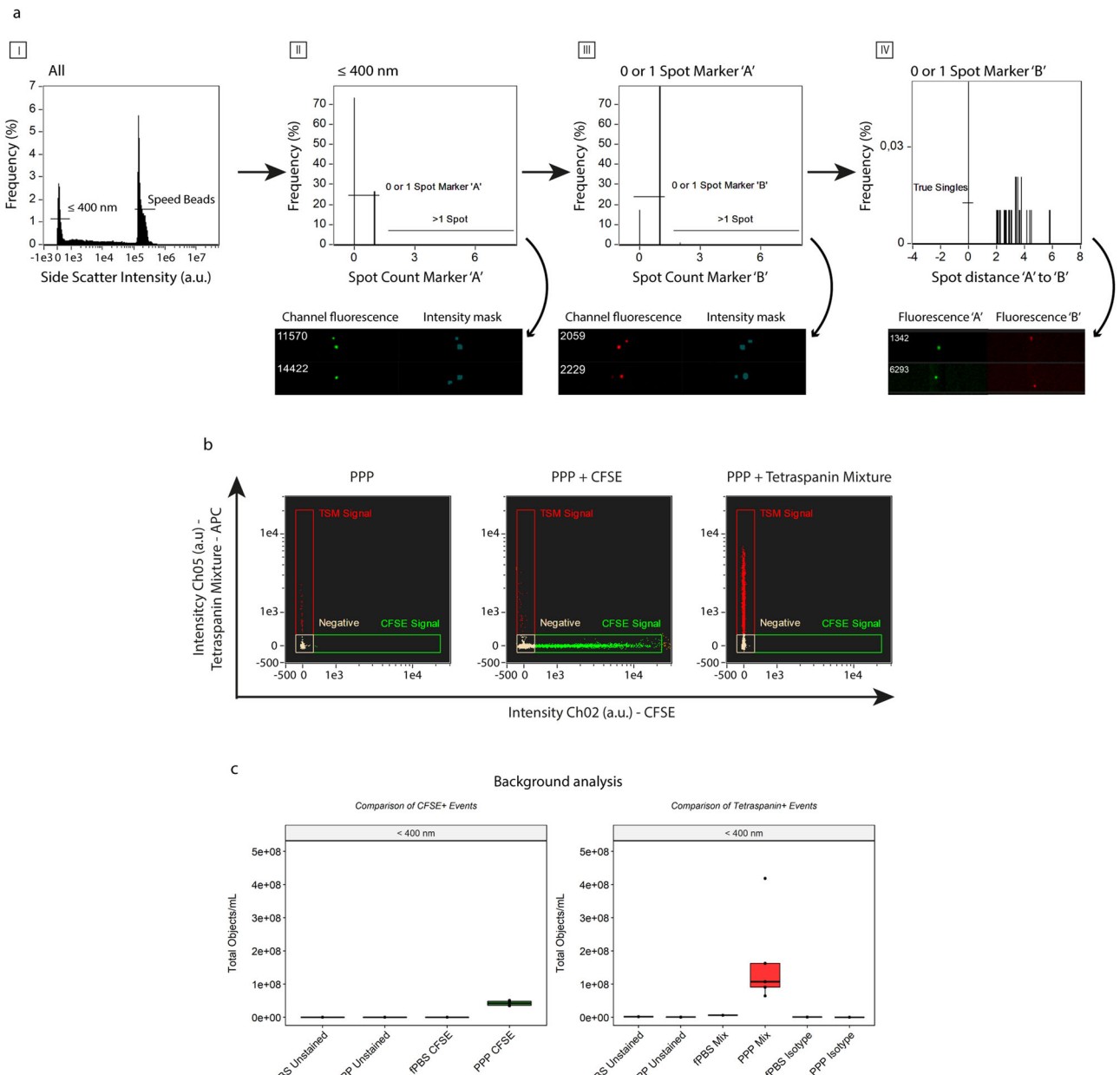

**Fig. 2 Gating strategy for the detection of single EVs through exclusion of coincident events. a** Generalized concept. First, particles with SSC intensities ≤ 900 a.u. are selected, effectively selecting all (fluorescent) particles ≤400 nm (I). Subsequently, coincidence detection is carried out based on the number of fluorescent spots within the pixel grid determined with the standard intensity mask. Events showing 0 or 1 spot within each channel are selected and used in the subsequent analysis (II & III); events showing more than 1 spot are excluded from analysis. Lastly, the distance between the individual fluorescent spots on the different detection channels is calculated and events not overlapping on the pixel grid are excluded (IV). Visual examples of excluded events are shown below each graph. **b** Representative example of unstained and single-stained PPP samples (stained with CFDA-SE or the anti-tetraspanin mixture -composed of anti-CD9/anti-CD63/anti-CD81-APC) used in the setting of the gating areas and identification of fluorescent events. X-axis: fluorescence intensity of CFSE, detected in channel 2 (Ch02). Y-axis: fluorescent intensity of the anti-tetraspanin mixture detected in channel 5 (Ch05). **c** Background analysis of fluorescent events (left: CFSE, right: anti-tetraspanin mixture) for unstained fPBS (Buffer Control), 5 unstained PPP, 1 single-stained fPBS and 5 single-stained PPP. Black dots: individual PPP samples.

and combined this new mask with the "Min Spot Distance" feature to calculate the distance between the fluorescent spots across the detection channels used. We then excluded all fluorescent events that did not occupy the same location on the pixel grid (Fig. 2a–IV). Ultimately, this gating strategy allows for the identification and subsequent analysis of single fluorescent submicron-sized particles ≤400 nm in PPP and is applied throughout the rest of this work.

**Establishment of IFCM background fluorescence.** Given their physical characteristics, EVs yield faint fluorescent signals – compared too cells – when measured with IFCM. Therefore, we assessed the fluorescent background levels induced by our staining protocol. As no washing steps are performed, the discrimination of EVs from fluorescent background signals is required to exclude false-positive particles from analysis. 0.20 μm filtered PBS (fPBS - Buffer Control) and platelet-poor plasma

(PPP) samples from 5 healthy individuals was stained with CFDA-SE (carboxyfluorescein diacetate succiminidyl ester) or a mixture of tetraspanin-specific antibodies (anti-CD9/anti-CD63/anti-CD81) labeled with APC. CFDA-SE is a non-fluorescent molecule converted to fluorescent CFSE (carboxyfluorescein succinimidyl ester) by intravesicular esterases. This helps to discriminate EV from lipoproteins, as the latter do not contain esterase activity.

PPP samples left unstained or singly stained with CFSE (Ch02) or the tetraspanin-specific antibody mixture (Ch05) were used to set the gating areas (Fig. 2b) and compensation matrix. Following our gating strategy, analysis of unstained fPBS or unstained PPP or fPBS + CFSE resulted in ~$E^5$ single-positive objects/mL within the CFSE gating area. In contrast, PPP samples single stained with CFSE showed an average of $4.23E^7 \pm 7.28E^6$ objects/mL (mean ± standard deviation), representing a 100-fold higher CFSE single-positive particle concentration compared to the unstained samples and fPBS (Fig. 2c, left panel).

Similarly, analysis of positive fluorescent events upon staining with the tetraspanin-specific antibody mixture showed that fPBS + anti-tetraspanin antibodies (fPBS Mix) yielded $5.98E^6$ objects/mL – a 3.6-fold increase over the concentrations of fPBS Unstained ($1.65E^6$ objects/mL). Additionally, an isotype control was added to analyze the specificity of the antibodies in the tetraspanin mixture. Positive particle concentrations were obtained for both fPBS and PPP Isotypes, ($6.16E^5$ and $1.97E^5 \pm 1.07E^5$ objects/mL, respectively). Analysis of PPP + anti-tetraspanin antibodies (PPP Mix) revealed an average of $1.69E^8 \pm 1.44E^8$ objects/mL – a 28-fold higher particle concentration than fPBS + anti-tetraspanin antibodies, a 350-fold higher particle concentration than PPP Unstained ($4.86E^5 \pm 2.6E^5$ objects/mL), and an approximate 860-fold higher particle concentration than PPP Isotypes (Fig. 2c, right panel). An approximate 4-fold higher concentration of fluorescent particles was observed in the PPP Mix vs CFSE after subtraction of background concentrations before comparison.

Together, these findings show that positive fluorescently stained events can be successfully discriminated from background signals and that the anti-tetraspanin antibody binding in our protocol is specific. Moreover, as unstained samples and isotype controls yielded ~$E^5$ (for CFSE) and fPBS with anti-tetraspanin antibodies yielded ~$E^6$ objects/mL in their respective fluorescent channels, we established the level of the background concentrations in our setup for single positive fluorescent events at $E^5$ and $E^6$ objects/mL, for CFSE ant anti-tetraspanin antibodies respectively.

**Human plasma single EVs can be discriminated from artifact signals through detergent treatment**. After optimizing the protocol to identify single fluorescent submicron-sized particles above background in PPP of healthy individuals, we tested the protocols' ability to discriminate legitimate EV signals from artefact signals. We hypothesized that single EVs could be identified as double-positive events after staining with both CFDA-SE and the anti-tetraspanin mixture, as these events would represent structurally intact, esterase-containing submicron sized particles bearing common EV antigens. To test this hypothesis, we examined the fluorescent populations of particles ≤400 nm in diameter in 1 fPBS and the same 5 PPP samples by combining both fluorescent stains. Following our gating strategy, gating areas were re-established on the basis of unstained and single-stained fPBS and PPP samples, as well as isotype controls. Gating cut-offs were determined to encompass all obtained fluorescent events for all PPP samples. Visual interrogation of the events in the identified fluorescent gates confirmed that the events analyzed met the

criteria imposed by the gating strategy: (co-localized) single-spot fluorescence (Fig. 3a).

After acquisition of double-stained PPP (Fig. 3b–I), we used detergent treatment (30 minutes incubation with 20 μL 10% (v/v) TritonX-100) to disrupt the lipid bilayer of EVs and thereby remove EV signals from the measurement (Fig. 3b–II). Fluorescent particles such as free antibodies or disrupted membrane fragments bearing antigens-antibodies remaining after detergent treatment were measured to allow the identification of artifact events, and the number of fluorescent events still present after detergent treatment were compared with the number of total fluorescent events before detergent treatment on a gate-by-gate basis to identify false-positive signals (Fig. 3c–e).

Analysis of CFSE single-positive events before detergent treatment showed a total of $3.25E^7 \pm 1.16E^6$ objects/mL acquired for PPP samples, and a 31% reduction was observed after detergent treatment resulting in $2.25E^7 \pm 1.03E^6$ objects/mL (~69% of total CFSE-single positive fluorescent events) (Fig. 3c).

Analysis of antibody mixture single-positive events showed a total of $1.47E^8 \pm 9.35E^7$ objects/mL events acquired for PPP samples, and $5.31E^7 \pm 6.88E^7$ objects/mL after detergent treatment (~36% of total events (Fig. 3d).

Analysis of double-positive events revealed $5.96E^7 \pm 3.69E^7$ objects/mL total double-positive particles across the 5 PPP samples measured, with a very limited number of artifact particles present after detergent treatment: $3.47E^6 \pm 4.48E^6$ objects/mL (~6% of total acquired events). This revealed that almost all double-positive particles measured ($5.61E^7 \pm 3.36E^7$ objects/mL, ~94% of the total concentration before detergent treatment), were structurally intact, esterase-containing EVs displaying common EV protein signatures in the form of tetraspanin markers (Fig. 3e).

By treating our samples with detergent we were able to identify to what extend our protocol discriminates legit EV signals from artefact signals. We observed that double-positive events were largely comprised of true EVs whereas the single-positive populations showed a high degree of fluorescent particles still present after detergent treatment. Therefore, we concluded that the colocalization of two fluorophores (found as double-positive events before detergent treatment) represent CFSE + /Tetraspanin+ EV.

**Fluorescent calibration for standardized reporting**. As mentioned before, flow cytometers differ in their fluorescent sensitivity and dynamic range, and therefore data comparison between different instruments is hindered. In order to improve data comparison fluorescent calibration must be performed to convert arbitrary units (a.u.) into standardized units. To this end, we used commercially available Rainbow Calibration Particles (RCP) with known reference values in terms of the Equivalent number of Reference Fluorophores (ERF).

Using the same settings as applied for EV measurements, we measured the Mean Fluorescent Intensity (MFI) of each of the four RCP bead populations (1 blanc – 3 fluorescent) generated for each channel used in our setup (Fig. 4a). Using the blank bead populations, we established the lower detection thresholds for fluorescent detection channels Ch02 (CFSE) and Ch05 (APC). We then calculated the respective logarithmic values of each peak (Fig. 4b), and performed a linear regression analysis of the ERF values against the MFI for peaks 2 to 4, omitting the blanc beads as these represent PS beads without fluorophores (Fig. 4c). In the example of the double-stained PPP sample presented in Fig. 3b without fluorescent calibration, we next converted the measured fluorescent intensities for CFSE and APC of each event into their respective ERF values (Fig. 4d). Lower fluorescent thresholds were

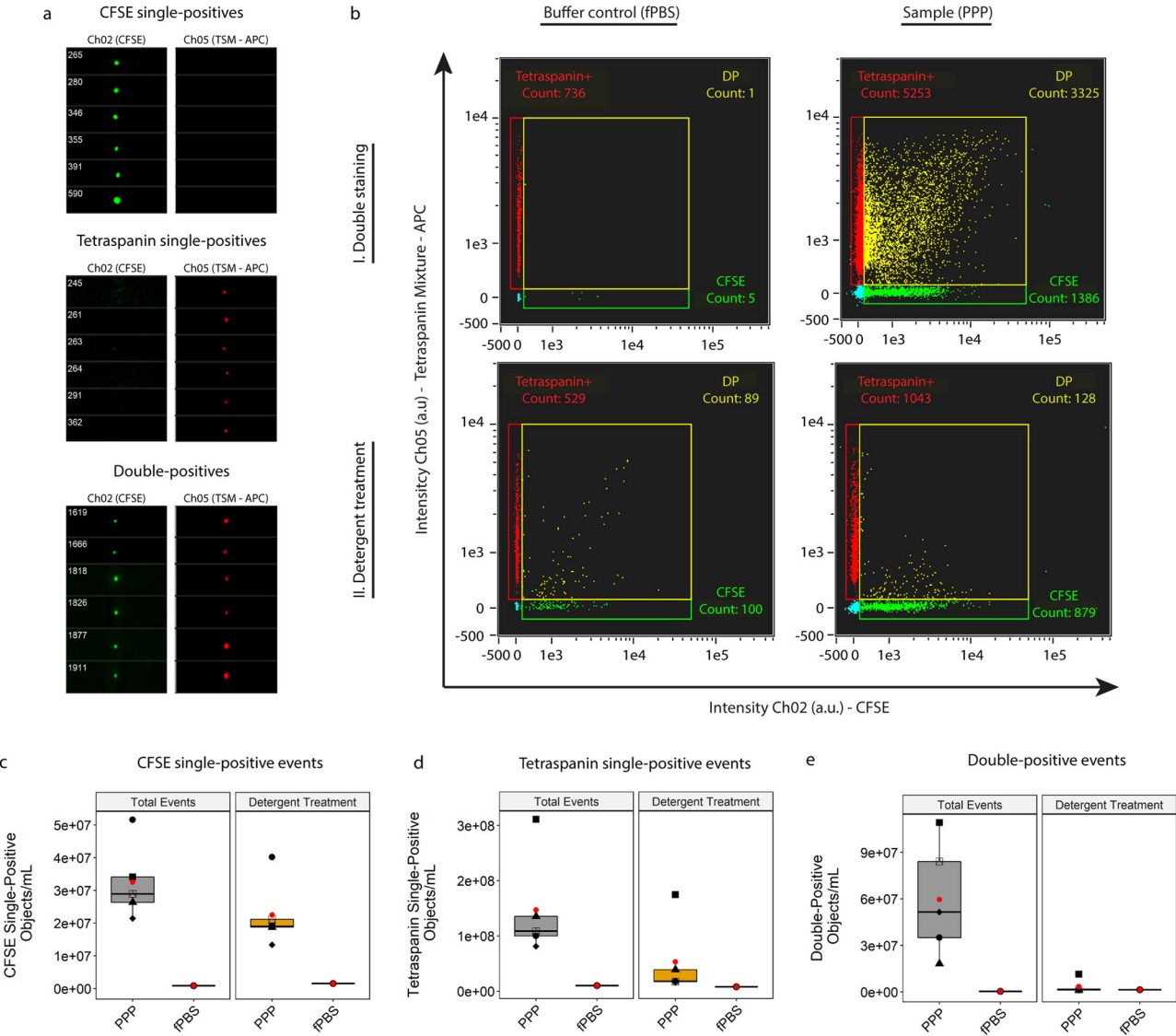

**Fig. 3 Identification of true EVs in PPP. a** Images of representative CFSE single-positive, tetraspanin single-positive and double-positive particles obtained from a double-stained PPP sample before detergent treatment. **b** Double-stained (CFDA-SE & anti-tetraspanin mixture) fPBS or PPP without (I) and with (II) detergent treatment to determine potential artifact signals. Detergent treatment was performed by incubating the samples for 30 minutes with 20 μL 10% (v/v) TritonX-100 stock solution. Analysis of CFSE single-positive **c**), Tetraspanin single-positive **d**) and double-positive fluorescent events **e**) in 5 PPP samples and fPBS before and after detergent treatment (gray and orange boxes, respectively) to discriminate true EVs from artifact signals on a gate-by-gate basis. Double-positive events were found to represent mostly true EV signals (~94% of total acquired double-positive events). Red dots: means of sample spread. Symbols: individual PPP samples.

converted accordingly and resulted in 35.40 and 6.40 ERF for CFSE and APC, respectively. Upper fluorescent thresholds were calculated at 3776 and 123 ERF for CFSE and APC, respectively. For the double-positive fluorescent population, this conversion resulted in median values of 138.09 ERF CFSE and 27.88 ERF APC.

These data show that the fluorescent intensities generated by imaging flow cytometry can be readily converted into standardized units, which, in turn, enhances the comparability of the generated data with other instruments using the same filter sets.

**Testing EV coincidence occurrence through serial dilution.** The detection of multiple EVs as a single event can lead to false interpretation of the data (e.g. underestimation of the concentration of particles of interest). To examine the accuracy of quantification of EVs from PPP by our IFCM protocol, we double

stained the 5 PPP samples with CFDA-SE and the anti-tetraspanin antibody mixture and performed a serial dilution experiment. The concentrations and ERF of double-positive particles in each PPP sample obtained after four 4-fold dilution steps were analyzed using a linear regression model, with the results shown in Fig. 5. All data shown were used in the analysis and $R^2$ calculation.

We observed that the concentrations of double-positive events were linearly proportional to the dilution factor (Fig. 5a) while the ERF of both fluorescent signals remained stable: mean 113.47 (range 55.07–157.55) for CFSE and mean 31.83 (range 28.2–36.8) for APC (Fig. 5b), showing that the IFCM platform is capable of accurately quantifying individual EVs. Serial dilution resulted into a larger spread of CFSE ERF values at lower dilutions (64x and 256x) only, which was interpreted to be a consequence of the lower number of particles analyzed. Additionally, double-positive EV concentrations at the aforementioned dilutions

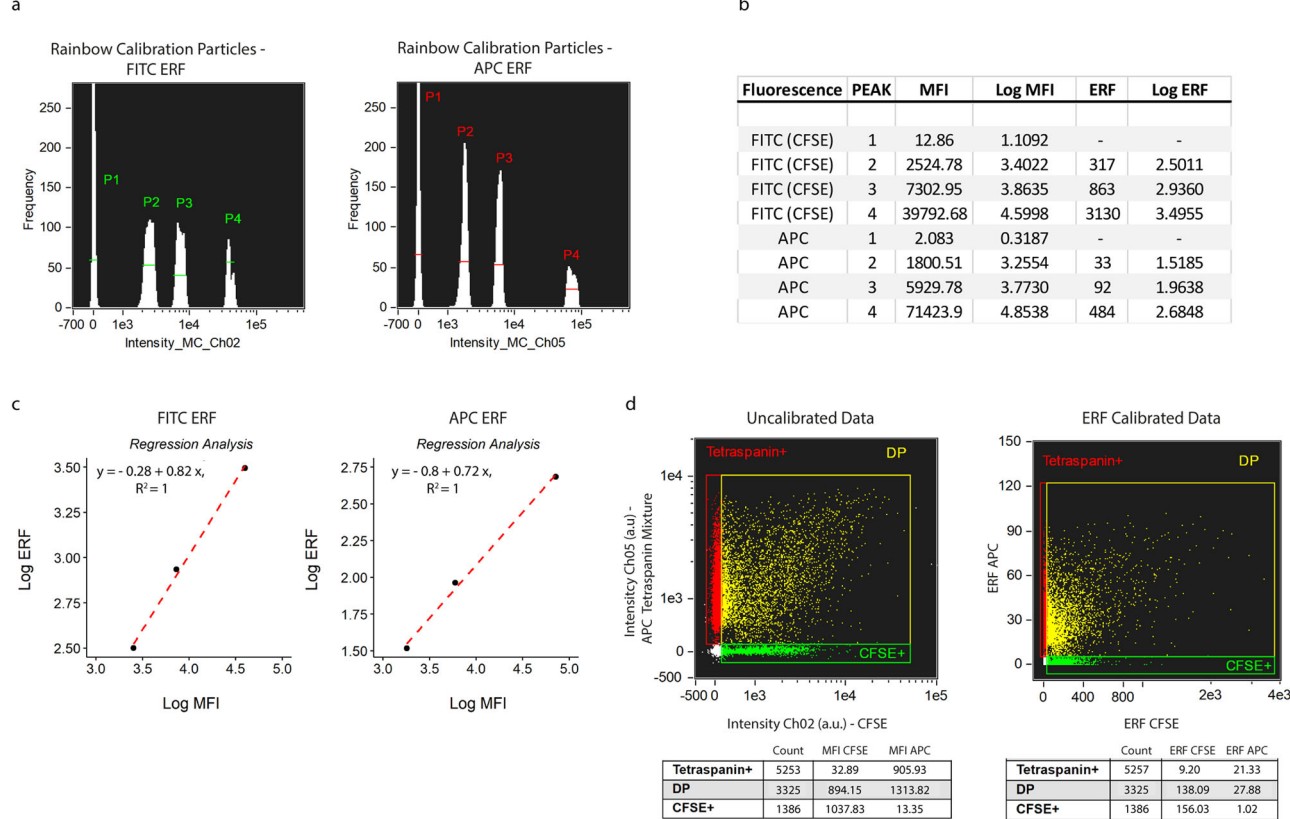

**Fig. 4 Fluorescent calibration allows reporting of fluorescent intensities in standardized units. a** The median fluorescent intensities (MFI) of each peak of FITC and APC ERF (Equivalent number of Reference Fluorophores) calibration beads was measured with the same instrument/acquisition settings applied as used for EV acquisition. **b** Calculation of the log of the MFI and ERF values (provided by the bead manufacturer). **c** For each of the used detection channels, the log of the MFI corresponding to the fluorescent peaks (P2-P4) was plotted on the x-axis, and the log of the ERF values on the y-axis; linear regression analysis was performed. **d** Representative example of uncalibrated data (left) and corresponding ERF calibrated data (right).

came close to the previously established background of our assay ($\sim E^5$ objects/mL).

The observed linear reduction in the concentration of double-positive events according to the dilution factor, and the stable ERF signals for both fluorescent markers, confirm that the IFCM platform is able to quantify true single EVs. Additionally, we were able to verify that our gating strategy correctly identifies and selects single EVs (by excluding multiplet events).

**Tetraspanin distribution on human plasma-derived single EVs.** After having established that our IFCM methodology identifies and quantifies single EV through staining with CFDA-SE and the anti-tetraspanin antibody mixture, we aimed to analyze whether we could detect different subsets of EVs. Therefore, we assessed the contributions of the individual tetraspanins to the double-positive events pool. The 5 PPP samples were stained with CFDA-SE and either the anti-tetraspanin antibody mixture or one of its individual components (anti-CD9 [clone HI9a], anti-CD63 [clone H5C5] or anti-CD81 [clone 5A6]) at a concentration equal to that used within the mixture. The concentrations of double-positive events upon staining with each stain were compared (Fig. 6a) and normalized with respect to the concentration of double-positive events (in objects/mL) obtained with the anti-tetraspanin antibody mixture (Fig. 6b).

The tetraspanin marker CD9 was found to be the main contributor to the fluorescent signal and thus responsible for most of the double-positive EVs identified in PPP when stained with the anti-tetraspanin antibody mixture: $\sim 88 \pm 11\%$ of the double-positive events were still present when staining with only

CD9 versus $\sim 13 \pm 3\%$ for CD63 and $\sim 9 \pm 5\%$ for CD81. In short, we show that our methodology is able to identify subsets of EVs, and that tetraspanin marker CD9 – and not CD63 or CD81 - represent the bulk of CFSE + single EVs in PPP of healthy individuals.

**Colocalization of fluorophores indicates true EVs.** So far, the identification and discrimination of single EVs from contaminating agents such as lipoproteins in PPP samples has been based on the notion that lipoproteins do not contain esterases, and hence cannot become fluorescently labelled by CFSE. However, as not all EVs may contain esterases the quantification of double-positive events (CFSE + /Tetraspanin + ) likely represents an underrepresentation when it comes to total EVs. An alternative approach to the identification of single EVs in PPP samples on the basis of intravesicular esterases would be the staining of samples with monoclonal antibodies (mAbs) targeting EV surface proteins. Based on the results presented in Fig. 6b, we used anti-CD9 [clone HI9a] as this antibody was shown to recapitulate the majority of the tetraspanin signal. Anti-CD31 [clone WM-59] was chosen as a secondary marker since CD31 is ubiquitously expressed within the vasculature and on diverse immune cell types, and therefore likely to be highly prevalent on EVs in PPP.

Figure 7a shows the ERF calibrated (APC calibration performed as described in Fig. 4, for BV421 calibration see Supplementary Fig. 1) IFCM results after double staining of both fPBS and a representative PPP sample with anti-CD9-APC and anti-CD31-BV421 and subsequent detergent treatment. The lower fluorescent threshold for Ch01 (BV421) was established

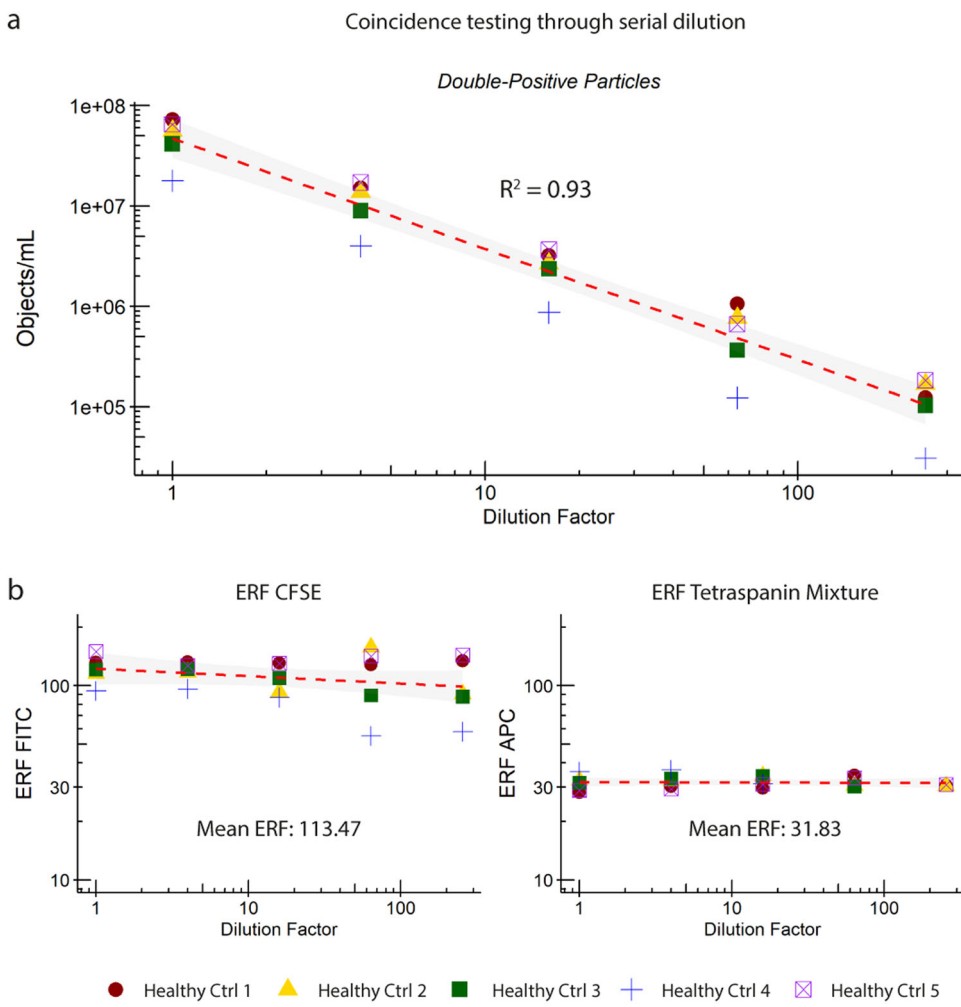

**Fig. 5 Examination of the accurate quantification of single EV detection by IFCM.** Analysis of serial dilutions of 5 double-stained (CFDA-SE & anti-tetraspanin mixture) PPP samples showed a linear correlation between (**a**) the obtained concentration and (**b**) Equivalent number of Reference Fluorophores (ERF) of fluorescent detection channels Ch02 (CFSE) and Ch05 (APC) with dilution factor (4-fold).

at 677.71 ERF; upper fluorescent threshold was established at 112,201 ERF. A visual representation of the events before detergent treatment within each gate is shown in Fig. 7b. As stated before, only single spot fluorescent events (with colocalized fluorescent spots for double-positive events) were analyzed.

Focusing on double-positive particles, we acquired a total of $5.12E^7 \pm 1.02E^7$ objects/mL before detergent treatment and $3.61E^6 \pm 5.46E^6$ objects/mL (~7% of total events) after detergent treatment, thus showing that ~93% of the double-positive events detected in the PPP sample could be classified as true single EVs with this strategy. Mean ERF values of the double-positive events in all 5 PPP samples (before detergent treatment) were calculated at ~7620 (range 3640–9240) and 20.4 (range 15–27.9) for BV421 and APC, respectively. Additionally, analysis of fPBS + mAbs (both anti-CD9 and anti-CD31 antibodies), PPP + isotype controls and fPBS + isotope controls yielded particle concentrations within the previously established fluorescent background range (~$E^5$ objects/mL), both before and after detergent treatment - indicating that the double-positive single EVs detected in the PPP + mAb samples were detected well above the level of the fluorescent background concentrations (Fig. 7c).

Thus, the staining of PPP samples with anti-CD9 and anti-CD31 showed that double-positive events (before detergent treatment) can be successfully identified as true single EVs. Although this staining approach (the combination of two surface

markers expressed on EVs) differs from the previously used staining approach (the combination of CFDA-SE and the anti-tetraspanin antibody mixture), both strategies resulted in the identification of true EVs on the basis of the colocalization of two fluorophores within the same event – indicating that this colocalization is membrane facilitated and therefore can be used as criteria to identify EVs in unprocessed PPP.

**IFCM facilitates specific EV subset analysis in contaminated/ diluted PPP samples.** To demonstrate the discriminative capabilities of our methodology, and to show that our staining procedure is specific, we mixed human and mouse PPP at various ratios (10% increments) and stained these samples with CFDA-SE and both anti-human CD31-BV421 and anti-mouse CD31-APC mAbs. For the analysis, all CFSE-positive events <400 nm were selected, and human and mouse single EVs were identified based on the species-specific antibody, thus ensuring the analysis of double-positive events.

Quantification of total human and mouse single EVs in 100% human or mouse PPP revealed a ~13-fold higher concentration in human: $2.29E^7 \pm 6.25E^6$ (CFSE + anti-human CD31 +, Fig. 8a) vs $1.8E^6 \pm 3.46E^5$ (CFSE + anti-mouse CD31 +, Fig. 8b) objects/mL, respectively. As expected, human EV concentrations showed a linear increase as the fraction of human PPP increased

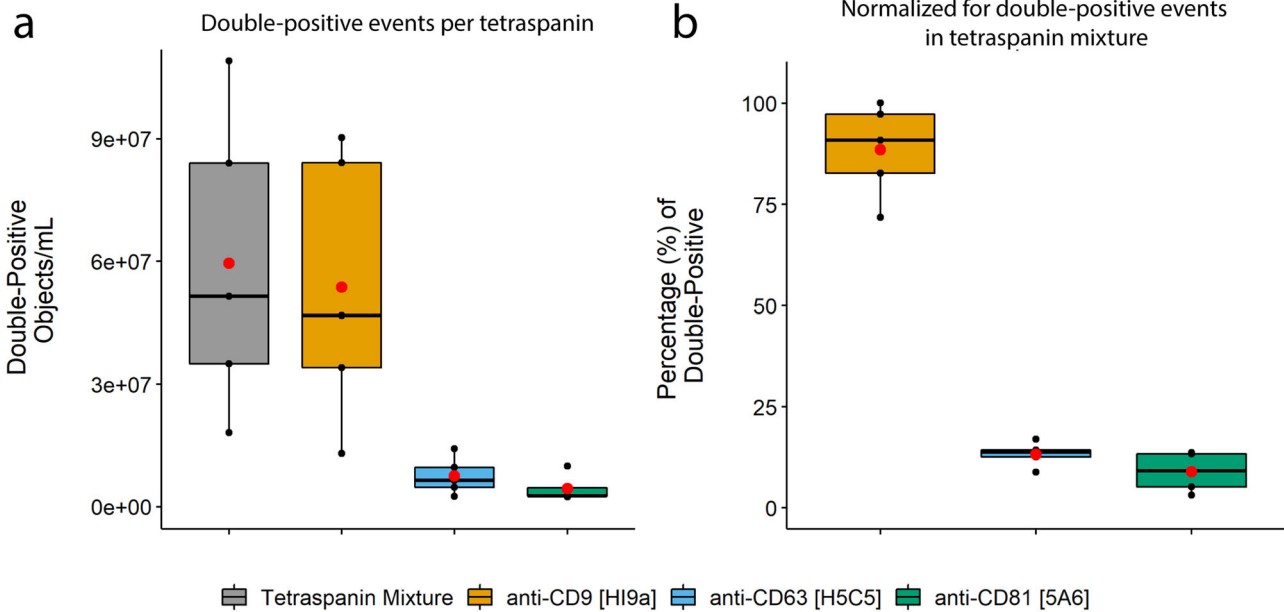

**Fig. 6 Tetraspanin distribution within 5 PPP samples.** All samples were stained with CFDA-SE and an anti-tetraspanin mixture or one of the anti-tetraspanin antibodies at a concentration equal to that used in the mixture. **a** Tetraspanin distribution determined using anti-CD9 [HI9a], anti-CD63 [H5C5] and anti-CD81 [5A6], and (**b**) their relative frequencies of double-positive events compared to that obtained with the anti-tetraspanin mixture. Results shown represent events (double-positive objects/mL) obtained with each of these staining combinations and are colored as follows: gray boxes – anti-tetraspanin mixture, orange boxes – anti-CD9, blue boxes – anti-CD63, green boxes – anti-CD81. Red dots: means of sample spread. Black dots, individual PPP samples.

($R^2 = 0.95$), while mouse EV concentrations showed the opposite trend (linear decreased as the fraction of human PPP increased – $R^2$ 0.81). Anti-human and anti-mouse concentrations obtained after staining the 100% human and mouse samples with their corresponding isotype controls were used to establish the background concentrations of our protocol (as indicated by the dashed red lines in Fig. 8a, b), and showed that the detection of anti-human/mouse EV is specific and above background. Additionally, no mAb cross-reactivity between species was observed.

Together, these data show that our method enables the discrimination and accurate quantification of distinct single EV populations in unprocessed, mixed PPP samples.

## Discussion

We developed an IFCM-based methodology to identify, phenotype and determine the concentration of single EVs in molecular complex blood plasma without prior isolation, providing an advantage over currently available analytical techniques, which *do* require EV isolation. We present an easy-to-use sample processing and staining protocol, and provide a gating strategy for the identification of single EVs. Following this gating strategy, EV subpopulations in PPP could be readily discerned based on the colocalization of two fluorescent markers bound to EV membranes. Additionally, platform standardization through both size and fluorescence calibration allows reproducibility and comparison of acquired data, showing the potential of our method for translation into clinical application.

Given that neither the isolation of EVs from PPP nor sample washing after staining with fluorescently labelled mAbs was performed, it was imperative to assess the fluorescent background levels induced by our sample handling protocol. Using control samples, we showed minimal background fluorescence and clear discrimination of specific fluorescent events above background. Approaches taken by other groups analyzing EVs in PPP using IFCM involve sample isolation[30] and/or washing steps to remove

unbound mAbs[26,30]; here we show that such sample isolation and/or washing steps can be omitted by detecting and eliminating the background produced by samples. We established the background level of the IFCM with respect to sub-micron particle quantification at ~$E^5$ objects/mL (after sample dilution correction). Previously published work by Görgens et al. showed that IFCM is able to accurately quantify single EVs (cell culture-derived) up to concentrations of ~$E^8$ objects/mL[21]. Together, these data suggest that single EV quantification with IFCM is optimal for samples between $E^5$ – $E^8$ objects/mL (as demonstrated in this work).

To identify single EVs present in the PPP samples, we designed a gating strategy based on the imaging capabilities of IFCM. Several key features or advantages that contribute to IFCM being a more powerful platform for EV analysis compared to conventional FC include the slower flow rate, CCD-camera-based detection (enabling higher quantum efficiency compared to conventional photon multiplier tubes), and integration of detected signals over time using TDI[21]. Additionally, IFCM allows automatic triggering on all channels during acquisition, and thus EVs devoid of SSC signals may still be detected based on their fluorescent probes. Conversion of scatter intensities from arbitrary units into standardized units (using light scatter theory and Mie calculations[32]) enhances reproducibility across different FC platforms. By performing these calculations for the BF and SSC detector channels, we demonstrated that measured PS bead signal intensities in the BF channel did not correlate with the theorized model. Thus, although the BF channel has its merits for cell-based research, it should not be used for EV-based research. The high degree of correlation between predicted and measured scatter intensities ($R^2 = 0.91$) for the SSC detection channel underlines the utility of the SSC channel to relate scatter signals to standard units.

Both size and fluorescence calibrations are key in the validation of submicron-sized particle detection and reproducibility of the generated data, respectively[18]. In line with previously published

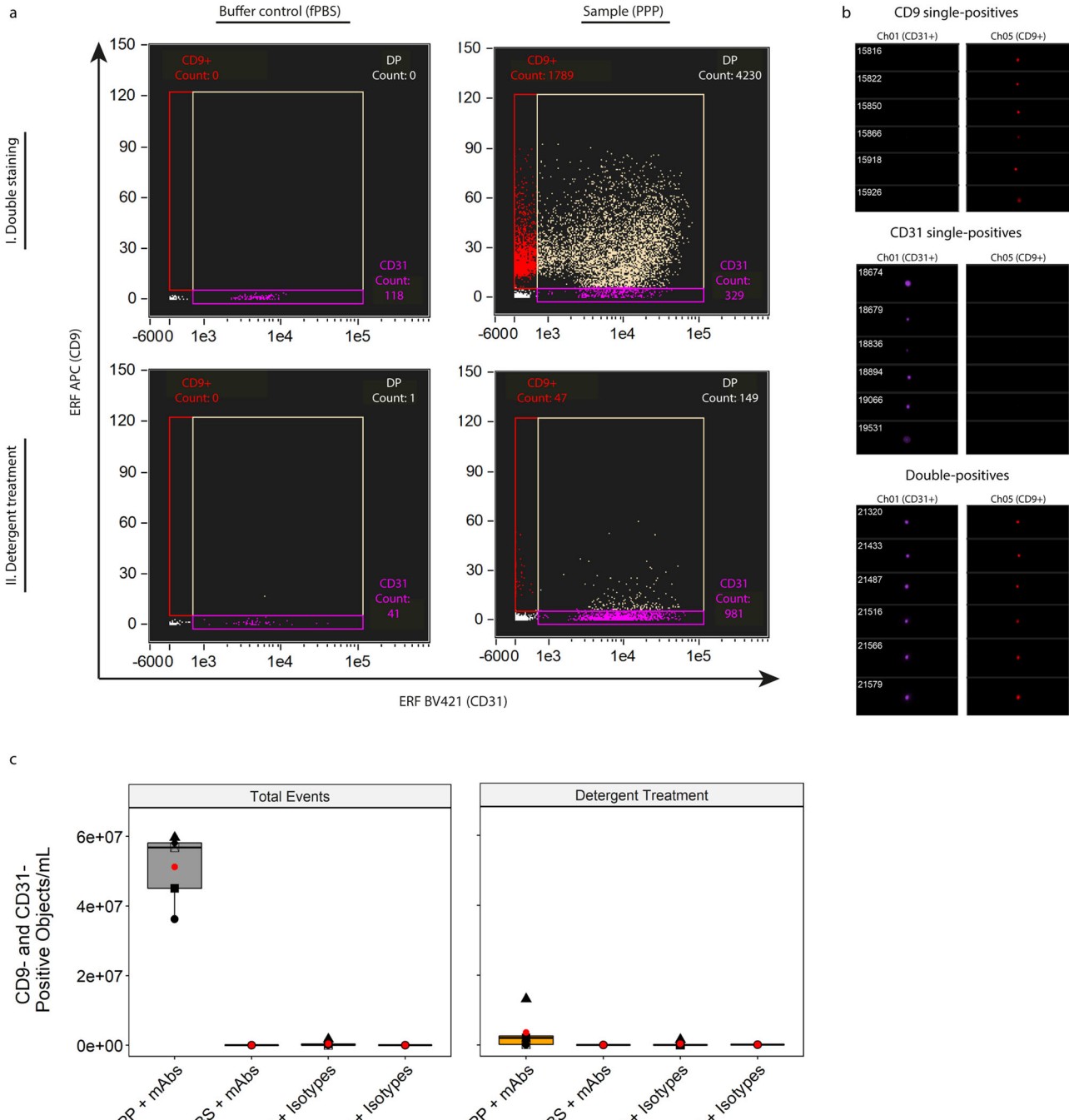

**Fig. 7 Identification of single EVs on the basis of vesicular surface markers. a** Representative, fluorescence calibrated data obtained for buffer control (fPBS, left column) and PPP (right column) samples stained with anti-CD31-BV421 and anti-CD9-APC mAbs. Detergent treatment was performed by incubating the samples for 30 minutes with 20 μL 10% (v/v) TritonX-100 stock solution. Red gate: Single-positive CD9 events, purple gate: single-positive CD31 events, tan gate: double-positive events. I, double staining and II, double staining after detergent treatment. **b** Visual interrogation of the gated populations in the representative PPP sample. **c** Quantification of double-positive fluorescent events in 5 PPP samples and fPBS, stained with mAbs or isotypes, before and after detergent treatment. Approximately 93% of double-positive events in PPP stained with mAbs represent PPP-derived single EVs that were detected well above the fluorescent background. Red dots: means of sample spread. Symbols: individual PPP samples.

literature, we have shown that IFCM is able to discriminate PS particles down to 100 nm on the basis of their emitted fluorescent intensities[21]. Regarding fluorescent calibration, we standardized the generated fluorescent intensities into ERF values using Rainbow Calibration Particles (RCP). It should be noted that ERF assignments to RCP are derived from a reference instrument, and comparisons across instruments are expected to vary with filter

and laser configuration, variations that can be measured and accounted for by cross-calibration against MESF or antibody capture beads[33].

A common artefact encountered when measuring sub-micron particles with conventional FC is swarm detection, which is defined as a special case of coincidence detection where instead of two or a few particles, multiple (tenths to hundreds) of particles at

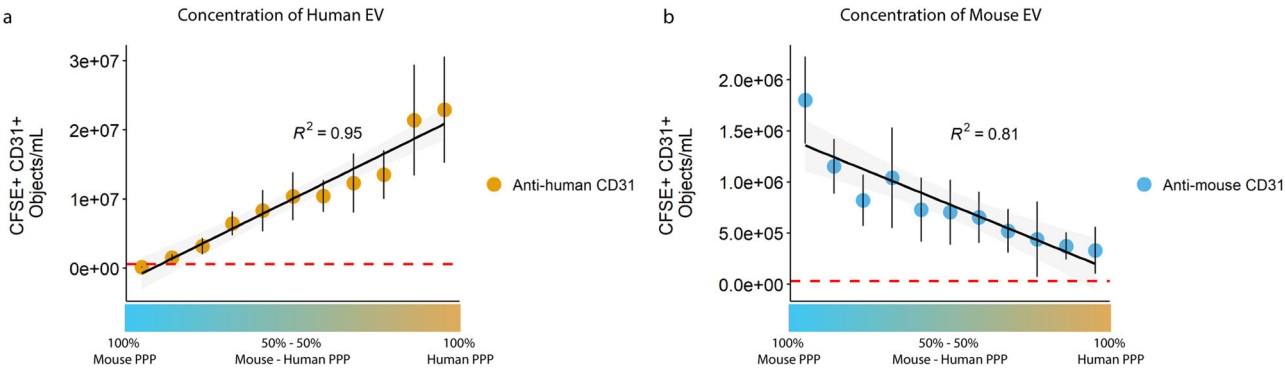

**Fig. 8 Quantification of single EVs in mixed human and mouse PPP samples.** Samples were stained with CFDA-SE and anti-human and anti-mouse CD31 (conjugated to BV421 and APC, respectively). Quantification of (**a**) CFSE + anti-human CD31 + single EV ≤ 400 nm showed a linear increase corresponding to the increase in human PPP abundance ($R^2 = 0.95$), while (**b**) CFSE + anti-mouse CD31 + single EV ≤ 400 nm showed a linear decrease corresponding to the decrease in mouse PPP abundance ($R^2 = 0.81$). Data were obtained through three independent experiments using the same human and mouse PPP samples. X-axis: v/v ratio of mouse – human PPP. Data shown represent the mean ± standard error. Red dashed lines: background concentrations of our protocol as indicated by the measurement of isotype controls.

or below the detection limit are simultaneously and continuously present in the laser beam of the flow cytometer and measured as single counts. This may occur during the detection of EVs in highly concentrated samples, and can lead to erroneous data interpretation[34]. While swarm detection can be prevented by dilution of highly concentrated samples, coincidence detection may still occur (albeit at lower frequencies). To identify coincidence detection, and exclude potential multiplets from our analysis, we designed a gating strategy that selects all events displaying 0 or 1 fluorescent spot on acquired images, thus ascertaining the analysis of events representative for single (and not multiple) particles. The identification of multiple, spatially separated fluorescent particles within acquired images provides insight into the degree of coincidence detection in a given sample - which is not possible with conventional FC. To demonstrate that our methodology correctly identifies and quantifies single EVs, we performed coincidence testing through serial dilution[35]. Analysis of the concentration of CFSE + /Tetraspanin+ EV upon serial dilution yielded a linear correlation with the dilution factor while ERF remained stable.

In this work we examined two fluorescent labeling strategies to identify and discriminate EVs: 1) application of CFDA-SE staining in conjunction with an anti-tetraspanin antibody mixture, and 2) staining with two mAbs targeting two different EV surface proteins. With both approaches, single EVs were identified through the colocalization of two fluorescent markers, thus excluding the possibility of soluble protein detection. The combination of isotype and detergent treatment controls demonstrated the specificity of the mAbs for EV labelling (and not lipoproteins), and the dissociation of lipid structures, respectively. Therefore, both of these controls are highly recommended, if not mandatory, for the correct interpretation of acquired results. Single EV concentrations as reported in this work are in line with concentrations reported by other groups obtained after the purification/isolation of PPP samples[29,36]. This shows the advantage of our methodology over existing analytical techniques as no isolation, and therefore less manipulation, of EVs are performed in our approach.

Another FC-based method to directly measure EVs in plasma, performed on a Beckman Coulter CytoFlex and using a strategy that encompasses the labelling of EVs with a fluorescent lipid probe (vFRed) in combination with CFDA-SE or an anti-tetraspanin mixture similar to ours, has recently been published[33]. In this study, membrane fluorescence was calibrated

in terms of vesicle size (surface area) by using a synthetic vesicle size standard, as provided in the vFC EV Analysis kit from Cellarcus Biosciences. However, the staining with a lipid membrane dye should be consistent for applicability. Thus, either the amount of dye needs to be approximately matched to the number of EVs, or an excess of dye should be used so that the membrane becomes saturated with dye[32]. Additionally, the staining of lipoproteins is unavoidable when performing lipid staining strategies on PPP samples.

It must be noted that the identification and quantification of single EVs through the IFCM method presented here is also subject to limitations. First, a minimum of 3 pixels is required before an event is recorded by IFCM as an object; fluorescent events not passing this threshold may consequently be missed. Second, our gating strategy excludes multiplets from analysis, and only single-spot fluorescent events are quantified. This may yield underestimations of EV concentration in very concentrated samples (as the frequency of multiplets may be higher than that of singlets during the acquisition of such samples)[21]. In such cases, serial dilution experiments may prove valuable to reduce multiplet detection and obtain a high frequency of single events. Alternatively, our gating strategy could be expanded upon: rather than excluding events representing multiplets, the individual particles might be quantified and – following multiplication of the obtained concentrations with a factor representing their identified multiplet value – added to the total obtained concentrations of singlets.

Combined, we propose five criteria for the successful analysis of single EVs in PPP through IFCM: 1) standardization of SSC signal intensities to allow estimation of particle sizes; 2) single-spot fluorescence to ensure single-particle analysis and no coincident events; 3) colocalization of a minimum of two fluorophores to assess the presence of two markers in the same particle or event; 4) disappearance after detergent treatment to confirm that the detected events represent structures composed of lipid membranes and hence are of biological origin; and 5) a linear correlation between concentration and dilution factor to further imply that single EVs are analyzed. These criteria are summarized in Table 1 for quick reference.

In conclusion, we present an IFCM-based methodology and provide a framework that will allow researchers to directly study plasma-derived EVs, expanding on the usage of EVs as non-invasive biomarkers in the clinic. We expect that this methodology, after validation of markers of interest, will be useful for

EV analysis in many different sample types and in a plethora of clinical settings.

## Methods

**Processing and storage of human blood plasma (steps I – III).** The collection and processing of samples from 5 healthy human individuals (2 males, 3 females, average age: 43.4 years, age range: 31–56 years) was approved by the Medical Ethical Review Board (MERB number MEC-2018-1623) and conducted in accordance with the Declaration of Helsinki. All individuals provided written informed consent. In brief, 12 mL of blood was collected (one drawing) from each individual into two BD Vacutainer® K3-EDTA-coated collection tubes (BD Biosciences, San Jose, USA) (Fig. 9–step I). Whole blood was centrifuged (Heraeus Multifuge 1 S) at 1910 × *g* for 10 min at room temperature (Fig. 9–step II). The plasma layer was then collected - leaving ~1 mm of plasma above the buffy coat - and centrifuged (Heraeus Fresco) at 16,000 × *g* for 10 minutes at room temperature in 1 mL aliquots using Safe-Lock Eppendorf tubes (Eppendorf AG, Hamburg, Germany). The resulting platelet-poor plasma (PPP) was first pooled before being divided into 700-µL aliquots in cryovials containing 28 µL of a 25x concentrated protease inhibitor cocktail solution (4% v/v) (cOmplete Protease inhibitor cocktail tablets, Roche, Mannheim, Germany) according to the manufacturers' instructions and stored at −80 °C (Fig. 9–step III).

**Processing and storage of mouse blood plasma.** All the procedures and animal housing conditions were carried out in strict accordance with current EU legislation on animal experimentation and were approved by the Institutional Committee for Animal Research (DEC protocol EMC No. AVD101002016635). Six weeks male C57BL/6J (JAX,GSP) mice (Jackson Labs, Bar Harbor, ME) were housed in the Erasmus MC animal facility and housed in groups of 2–3/cage. They were maintained on a 12:12 h light-dark cycle and allowed ad libitum access to water and standard rodent food. The mice were anesthetized and blood (approximately 0.8 mL) was collected via the left ventricle using a 23–25 gauge needle. To ensure euthanasia of the animal post-procedure, mice were killed by cervical dislocation.

**Antibody preparation (Step IV).** All monoclonal antibodies (mAbs) were centrifuged for 10 minutes at 16,000 × *g* to reduce the number of (potential) mAb clumps (Fig. 9 – step IV). A volume of the top layer of each centrifuged mAb solution was carefully harvested (according to the dilutions needed, described below) and diluted in 0.22 µm-filtered PBS (fPBS) before being added to the samples (Fig. 9 – step VI). The sample staining protocol is described under step VI. The mAbs used to stain human PPP were anti-CD9–APC, clone HI9a (6 µg/mL, BioLegend, San Diego, USA); anti-CD63–APC, clone H5C6 (200 µg/mL, BioLegend); and anti-CD81–APC, clone 5A6 (200 µg/mL, BioLegend. Human and mouse PPP were both stained with anti-human CD31–BV421, clone WM-59 (50 µg/mL, BioLegend) and anti-mouse CD31-APC, clone 390 (200 µg/mL, BioLegend). Isotype controls used were IgG1,k-BV421, clone MOPC-21 (100 µg/mL, BioLegend); IgG1,k-APC, clone MOPC-21 (200 µg/mL, BioLegend); and IgG2a,k-APC, clone RTK2758 (200 µg/mL, BioLegend). Optimal mAb concentrations were determined by performing separate titration experiments for each mAb on human

**Table 1 Criteria for events to be classified as true single EVs by IFCM.**

| # | Criteria | Reasoning |
|---|---|---|
| 1 | Standardization of SSC signals | Allows estimation of particle sizes |
| 2 | Single Spot Fluorescence | Single-particle analysis/no coincidence events |
| 3 | Colocalization of fluorophores | Indicating the presence of markers in the same particle/event |
| 4 | Signal disappears after Detergent Treatment | Confirmation that detected events are of biological origin |
| 5 | Linear correlation with Dilution factor | Single-particle analysis & confirmation that events are biological |

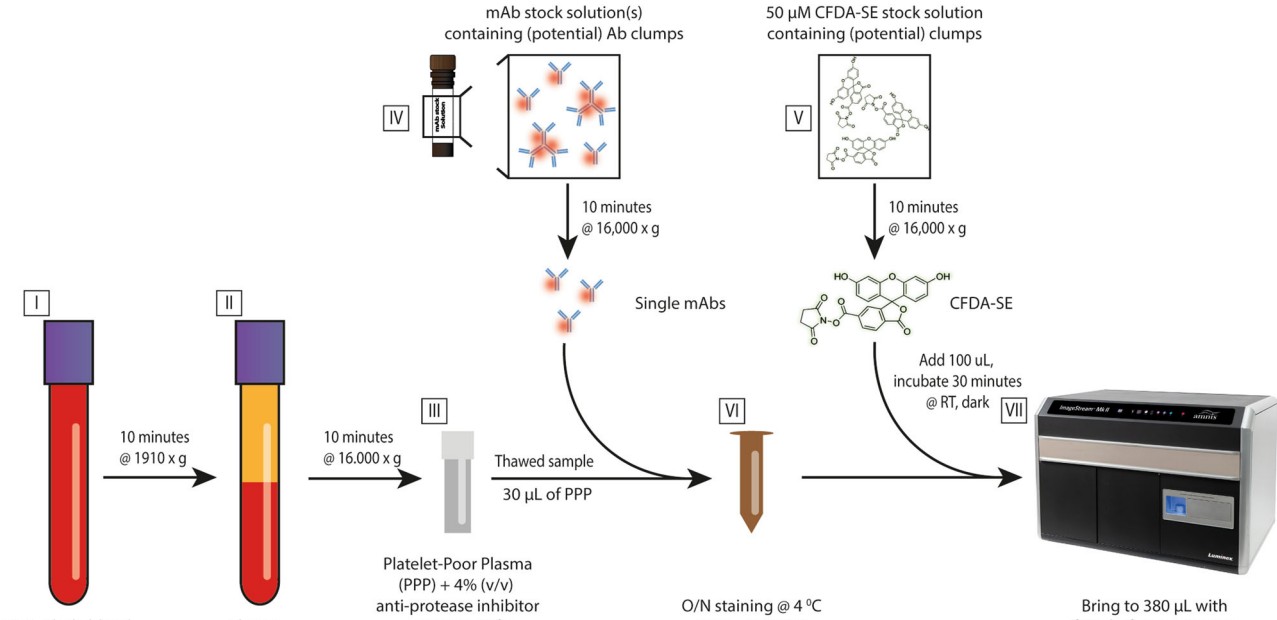

**Fig. 9 Schematic overview of the sample processing and staining protocol.** EDTA whole blood (I) is centrifuged for 10 min at 1910 × *g*, after which the plasma top layer (II) is harvested and subjected to centrifugation for 10 minutes at 16,000 × *g*. The resulting platelet-poor plasma (PPP) is collected and stored as aliquots in cryovials containing an anti-protease inhibitor stock solution (4% v/v) at −80 °C (III). Prior to use, monoclonal antibodies and CFDA-SE stock solutions are centrifuged for 10 min at 16,000 × *g* to reduce the number of potential clumps of fluorescent particles (IV & V). mAbs are added to 30 µL of PPP, and samples are brought to a total volume of 130 µL using filtered PBS (VI) and incubated overnight (O/N) at 4 °C in the dark. CFDA-SE is added on the day of acquisition and incubated for 30 minutes at room temperature in the dark. Samples are brought to a total volume of 380 µL using filtered PBS and are assessed using a Luminex ISx MKII imaging flow cytometer (VII). After initial acquisition, detergent treatment is performed for each sample by adding 20 µL 10% (v/v) TritonX-100 solution followed by 30 minutes incubation at room temperature. Permission to use the image of the Imagestream (ISx) Imaging Flow cytometer (VII) was granted by Luminex.

PPP and fPBS samples in parallel. The optimal concentration of each mAb was defined as the concentration that yielded the best discrimination between sample (PPP) and background (fPBS). All tetraspanin mAbs were diluted 30-fold in fPBS before staining (Final concentrations: CD9: 200 ng/mL, CD63: 6.6 μg/mL, CD81: 6.6 μg/mL); CD31-BV421 (antihuman) and CD31-APC (anti-mouse) were diluted 1000-fold (Final concentration: 50 ng/mL) and 62.5-fold (Final concentration: 3.2 μg/mL), respectively. The antitetraspanin antibody mixture was made by combining anti-CD9/anti-CD63/anti-CD81 in the same stock solution.

**Preparation of a carboxyfluorescein diacetate succinimidyl ester stock solution (Step V).** A carboxyfluorescein diacetate succinimidyl ester (CFDA-SE) stock solution was made with the Vybrant™ CFDA-SE Cell Tracer Kit from Invitrogen immediately prior to use according to the manufacturer's instructions. In brief, CFDA-SE powder was spun down using a table-top centrifuge, and 18 μL of dimethylsulfoxide (DMSO) was added. The mixture was thoroughly resuspended and incubated at room temperature for 10–15 min in the dark. Then, the dissolved CFDA-SE was added to a total volume of 1.782 mL of fPBS to create a 50 μM CFDA-SE stock solution. Similar to the protocol used to prepare mAbs, this stock solution was centrifuged for 10 minutes at 16,000 × g to reduce potential CFDA-SE clumps (Fig. 9 – step V); the top layer was carefully harvested – leaving ~100 μL of liquid in the tube – before being added to the samples.

**Sample labeling (Step VI).** Staining was performed overnight at 4 °C in the dark in a total volume of 130 μL. This volume was build-up by 30 μL of sample, a volume of mAb stock solutions (described under step IV) as needed and brought to the total volume of 130 μL with fPBS; 12.5 uL of the stock solutions containing mAbs labelled with –APC and 5 μL of the stock solutions containing mAbs labeled with –BV421 were added, resulting in the following concentrations used per test: CD9 – 2.5 ng, CD63 – 83 ng, CD81 – 83 ng, CD31 (antihuman) – 1 ng, CD31 (anti-mouse) –40 ng per test. Equivalent amounts of isotype control were used for each antibody. For specificity and sensitivity analysis, human and mouse PPP were mixed at varying ratios with the total volume of PPP maintained at 30 μL. Samples were then incubated overnight at 4 °C to ensure optimal saturation of the available EV epitopes (Fig. 9– step VI); this incubation time was determined empirically by adding the anti-tetraspanin antibody mix to fPBS and PPP samples and performing acquisition at set intervals (1/3/6 hours and O/N). CFDA-SE labeling was performed on the day of data acquisition by adding 100 μL of the 50 μM CFDA-SE stock solution to the samples, followed by 30 min of incubation at room temperature in the dark. Control samples not stained with CFDA-SE were incubated with 100 μL fPBS instead. All samples were brought to a total volume of 380 μL using fPBS before IFCM measurements.

**Controls.** Assay controls were used in all experiments, as recommended by the MIFlowCyt-EV framework[18] (Supplementary Tables 1 and 2). These controls consisted of fPBS, fPBS with reagents, unstained samples, single-stained samples, isotype controls (matched with their corresponding fluorophore-conjugated mAbs at the same concentrations) and samples subjected to detergent treatment. A 10% (v/v) Triton X-100 stock solution was made by dissolving 1 mL of TritonX-100 in 9 mL of fPBS. Detergent treatment was performed by the addition of 20 μL of the Triton X-100 stock solution (final concentration: 0.5% (v/v) per test), followed by 30 min of incubation at room temperature in the dark prior to acquisition. Note that samples were first acquired as described in the section "Data acquisition (Step VII)" before detergent treatment and corresponding re-acquisition was performed. Supplementary Table 3 gives an overview of these controls as well as the rationale behind their use. All controls contained 4% (v/v) 25x concentrated protease inhibitor cocktail solution (cOmplete Protease inhibitor cocktail tablets, Roche, Mannheim, Germany) in accordance with the PPP samples.

**Usage of polystyrene beads for calibration purposes.** A mix of commercial fluorescent polystyrene (PS) beads was used to calibrate fluorescence and light scattering signals. Megamix-Plus FSC (lot 203372) and Megamix-Plus SSC (lot 210812) beads (BioCytex) were mixed at a 1:1 ratio, resulting in a mix containing green fluorescent bead populations with sizes of 100, 300 and 900 nm from the Megamix-Plus FSC bead set, and 160, 200 and 240 nm from the Megamix-Plus SSC bead-set and 500 nm from both; this mix was termed Gigamix. Rainbow Calibration Particles (RCP-05-5, lot AL01, Spherotech) with known Equivalent number of Reference Fluorophores (ERF) values for C30/FITC/APC (as determined on a Beckman Coulter CytoFLEX) were used in the standardization of the fluorescent detection channels Ch01,Ch02 & Ch05, respectively. For each detection channel, the MFI of each peak from the four bead populations (1 blanc – 3 fluorescent) were measured, and a linear regression analysis was performed of the log of these values against the log of the known ERF values. The resulting linear function was used to relate the log of BV421/CFSE/APC fluorescent intensities to the log of ERF values.

**Light scatter theory and Mie calculations for IFCM.** Light scattering signals of bead populations from Gigamix were fitted with Mie theory using a previously described model[32]. The BF detector was modelled as a forward scattered light detector collecting light using a lens with a numerical aperture (NA) of 0.9, which corresponds to the NA of the 60x objective. The center wavelength of brightfield detection was 618.5 nm. The SSC detector was modelled as a detector that is placed perpendicular to the propagation direction of the laser beam. The NA of the collection lens was 0.9 and the wavelength was 785.0 nm. PS beads were modelled as solid spheres with a refractive index of 1.5885 for a wavelength of 618.5 nm (brightfield) and 1.5783 for a wavelength of 785.0 nm (SSC). EVs were modelled as core-shell particles with a core refractive index of 1.38, shell refractive index of 1.48 and a shell thickness of 6 nm for both wavelengths as the dispersion relation for the core and shell of EVs is unknown. Beads were measured in water, and EVs in PBS. Therefore, the refractive indices of PBS and water were assumed to be 1.3345 and 1.3325, respectively, at a wavelength of 618.5 nm (BF) and 1.3309 and 1.3289, respectively, at a wavelength of 785.0 nm (SSC).

Effective scattering cross sections of the calibration beads were calculated by integrating the amplitude scattering matrix elements over 576 collection angles[32]. Data and theory were log10-transformed to scale the data onto the theory using a least-square-fit.

**Data acquisition (Step VII).** All samples were analyzed on an ImageStreamX MKII instrument (ISx; Luminex, Texas, USA) equipped with 4 lasers set to the following powers: 405 nm: 120 mW, 488 nm: 200 mW, 642 nm: 150 mW, and 775 nm (SSC): 1.25 mW. The instrument calibration tool ASSIST® was used upon each startup to optimize performance and consistency. The ISx was equipped with three objectives (20x/40x/60x) and 1 CCD camera. Settings previously established by Görgens et al.[21] were used in our experiments. All data were acquired using the 60x objective (numerical aperture of 0.9 – wherein 1 pixel images an area of 0.1 μm²) with fluidics settings set to "low speed/high sensitivity" – resulting in a flow speed of 43.59 ± 0.07 mm/sec (mean ± standard deviation). We adjusted the default sample core size of 7 μm to 6 μm using the "Defaults Override" option within INSPIRE software (version 200.1.681.0), as recommended by the manufacturer. Data were acquired over 180 seconds for standardization among samples with the autofocus setting activated and the "Remove Speedbead" option unchecked. These settings are shown in Supplementary Table 4 for quick reference. BV421 fluorescence signals were collected in channel 1 (435–505-nm filter), CFSE signals in channel 2 (505–560-nm filter) and APC signals in channel 5 (642–745-nm filter). Channel 4 was used as the brightfield channel, and channel 6 (745–785-nm filter) was used for SSC detection. Particle enumeration was achieved through the advanced fluidic control of the ISx coupled with continuously running SBs (used by the IFCM to measure sample velocity for camera synchronization during acquisition, and enables particle enumeration during analysis), and application of the "objects/mL" feature within the ISx Data Exploration and Analysis Software (IDEAS®).

**Data analysis.** Data analysis was performed using Amnis IDEAS software (version 6.2). The image display mapping was linearly adjusted for all fluorescent events for each channel and then applied to all files from their respective experiments. The IDEAS software utilizes 'masks' – defined as the algorithm which selects pixels within an image based on their intensity and localization – to define the analysis area of each event within the pixel grid. The "masks combined" (MC) standard setting was used to quantify all fluorescence intensities in the channels used during acquisition corresponding to the fluorochromes used (Ch01, Ch02 & Ch05). Fluorescent events from singly stained PPP samples were used in the setting of compensation matrices (to compensate for spectral overlap between fluorochromes) such that straight fluorescent populations were obtained when depicted in scatterplots. Single-positive gating areas were established based on these single-positive fluorescent populations, and double-positive gates were set based on the boundaries of the single-positive gates. Unstained samples were used in the definition of the low-end of the various gates. Fluorescent thresholds were verified using cut-off values from the blanc fluorescent bead populations in the Rainbow Calibration Particles.

**Statistics and Reproducibility.** Statistical analysis was performed using R version 4.0.2 and RStudio (RStudio Team (2016). RStudio: Integrated Development for R. RStudio, Inc., Boston, MA; URL: http://www.rstudio.com/) version 1.1.463. All concentrations reported in this work were corrected for sample dilution (before acquisition – 380 μL total volume per test containing 30 μL sample = ~12.33-fold dilution factor) and are shown as the mean ± standard deviation unless specified otherwise. In all experiments conducted, PPP samples from the same 5 healthy individuals were used ($n = 5$ biologically independent samples). In the mouse vs human experiments, three independent experiments were conducted using the same mouse and human PPP samples (three replicates).

**Reporting summary.** Further information on research design is available in the Nature Research Reporting Summary linked to this article.

## Data availability

All source data underlying the figures presented in this work are provided as 'Supplementary Data 1–8' (separate tabs for each figure). Any other relevant data are available from the corresponding author upon reasonable request.

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

## Acknowledgements

The authors would like to thank Peter Rhein (Luminex) for his support and discussions throughout the project.

## Author contributions

W.W.W. participated in the research design, execution of the research, data analysis, and article drafting and is the corresponding author. E.vd.P. participated in the calibration of scatter intensities through Mie theory. E.M. participated in data acquisition and provided IFCM expertise. M.J.H., C.C.B., and K.B. participated in the research design. A.M. participated in the research design, data acquisition and data analysis. All authors reviewed the manuscript and approved its final version.

## Competing interests

The authors declare no competing interests.
