## [Peer Review File · Communications Biology]

REVIEWERS' COMMENTS:

Reviewer #1 (Remarks to the Author):

The revised manuscript by Woud et al. is another manuscript showing that single EV characterization works with IFCM. As the other reviewers pointed out, the manuscript is well written and shows well performed experiments, however the novelty of the projects is unclear to me. Multiple other groups have shown that IFCM works for EV characterization (Görgens, Ricklefs, Lanningan, etc.). The only real novelty is: IFCM + unprocessed plasma, which on the other hand has been shown for conventional Flow cytometry by multiple other groups. Therefore I strongly question the novelty of the manuscript.

REVIEWERS' COMMENTS:

Reviewer #1 (Remarks to the Author):

The revised manuscript by Woud et al. is another manuscript showing that single EV characterization works with IFCM. As the other reviewers pointed out, the manuscript is well written and shows well performed experiments, however the novelty of the projects is unclear to me. Multiple other groups have shown that IFCM works for EV characterization (Görgens, Ricklefs, Lanningan, etc.). The only real novelty is: IFCM + unprocessed plasma, which on the other hand has been shown for conventional Flow cytometry by multiple other groups. Therefore I strongly question the novelty of the manuscript.

Author reply: The current manuscript describes a comprehensive Imaging Flow Cytometry (IFCM)-based methodology for the *direct measurement* of Extracellular Vesicles in unprocessed human plasma. Plasma is easily obtained and represents an interesting matrix to search for both prognostic and diagnostic biomarkers in disease. To bring the EV biomarker research in human plasma a step further this manuscript serves as an important guideline.

Due to the instruments' internal configurations, IFCM is considered to be a more powerful platform for EV analysis compared to conventional FC: the slower flow rate, CCD-camera based detection (enabling higher quantum efficiency compared to conventional photon multiplier tubes), and integration of detected signals over time using TDI all contribute to an increased sensitivity (for small particle detection) over conventional FC. Consequently, whilst size calibrations following Mie theory have been shown for conventional FC, this manuscript is the first to show and apply the calibration of SSC signal intensities into particle size and scattering cross section (nm^2) for IFCM - which will advance this platform (and the EV field as a whole) in terms of single EV analysis. Furthermore, the step-by-step validation (following the MIFlowCyt-EV framework) of IFCM in this manuscript can be applied for the validation of other (future) FC platforms.

Taken together, this manuscript combines and presents the various recent advancements made in the field of EV Flow Cytometry and demonstrates for the first time its applicability in unprocessed human plasma. This will allow researchers to directly study plasma-derived EVs, expanding on the usage of EVs as non-invasive biomarkers in the clinic.